# Electrically Conducting Pullulan-Based Nanobiocomposites Using Carbon Nanotubes and TEMPO Cellulose Nanofibril

**DOI:** 10.3390/nano11030602

**Published:** 2021-02-28

**Authors:** Sabina Yeasmin, Jeong Hyun Yeum, Byung Chul Ji, Jin Hyun Choi, Seong Baek Yang

**Affiliations:** 1Department of Biofibers and Biomaterials Science, Kyungpook National University, Daegu 41566, Korea; yeasminsabina44@yahoo.com (S.Y.); jinhchoi@knu.ac.kr (J.H.C.); 2Department of Advanced Materials Science and Engineering, Kyungpook National University, Daegu 41566, Korea; bcji@knu.ac.kr

**Keywords:** pullulan, tempo cellulose nanofibrils, carbon nanotubes, nanocomposite film, thermal and mechanical properties, biodegradability, conductivity

## Abstract

Hybrid nanobiocomposite films are prepared using a solution casting by incorporating TEMPO cellulose nanofibrils (TOCNs) and carbon nanotubes (CNTs) into an aqueous solution of pullulan (PULL). The presence of CNT is confirmed by XRD characterization, and the prepared film shows an increased degree of crystallinity after the addition of TOCNs and CNT. The maximum degree of crystallinity value is obtained for CNT 0.5 % (59.64%). According to the Fourier-transform infrared spectroscopy, the shifts of the characteristic -OH peak of PULL occurred after the addition of TOCNs and aqueous CNT (3306.39 to 3246.90 cm^−1^), confirming interaction between the TOCNs, CNTs, and PULL matrix. The prepared films show enhanced material properties including higher tensile strength (65.41 MPa at low CNT content (0.5%)), water barrier properties, and reduced moisture susceptibility (5 wt.% CNT shows the lowest value (11.28%)) compared with the neat PULL film. Additionally, the prepared films are almost biodegradable within 64 days and show excellent electrical conductivity (0.001 to 0.015 S/mm for 0.5–5% CNT), which suggests a new approach to transform natural polymers into novel advanced materials for use in the fields of biosensing and electronics.

## 1. Introduction

Pullulan (PULL) is a well-known microbial exopolysaccharide, which is generated aerobically by specific strains of *Aureobasidium pullulans*, a polymorphic yeast [1]. This polysaccharide consists of maltotriose, which acts as a repeating structural unit [2]. Pullulan is highly soluble in water, possesses high adhesion, and has an exceptional fiber- and film-forming capability. Pullulan films are transparent, edible, harmless, thermally stable, and show anti-static and elastic properties, making them useful for packaging [3], food processing, and pharmacological applications [2]. Moreover, pullulan can be used after being chemically modified or blended with other polymers, revealing new functionalities, and, consequently, expanding its range of applications [4].

According to recent studies, nanocelluloses are divided into three main categories: cellulose nanocrystals or cellulose nanowhiskers, microfibrillated celluloses (MFCs), and cellulose nanofibrils. Cellulose nanocrystals or cellulose nanowhiskers are produced from native celluloses by acid hydrolysis followed by mechanical stirring of the acid-hydrolyzed filtrates in water [5]. MFCs are made by the mechanical breakdown of cellulose/water slurries with or without energy-reducing support by fractional carboxymethylation or cellulose treatment [6]. Cellulose nanofibrils are produced from native celluloses by 2,2,6,6-tetramethylpiperidine-1-oxy radical (TEMPO)-mediated oxidation with a subsequent mechanical breakdown of the oxidized celluloses in water [7]. In contrast to the first two categories of nanocellulose, TEMPO-oxidized cellulose nanofibrils (TOCNs), produced mainly from wood celluloses, have uniform widths of 3–4 nm, large aspect ratios of more than 50, and are dispersed as distinctive nanofibrils in water [8]. Recently, nanocellulose films have been widely studied with regard to their oxygen barrier properties for application in environmentally friendly films in packaging technology [9,10,11,12].

Carbon nanotubes (CNTs) have attracted attention due to their significant properties and applications [13,14,15,16,17,18,19,20,21,22]. However, CNT cannot be applied in its bulk form (e.g., aligned stacks, powder, films/papers, among others) because of the poor conversion of the excellent intrinsic properties of an individual CNT into its macroscopic appearances. Specifically, the integration of CNT as a filler material with different polymer-based matrices to obtain CNT/polymer nanocomposites [23] has revolutionized the field of materials science and technology. Consequently, the main applications of CNT are in its combination with other materials like blends, alloys, composites, or hybrid materials [24,25,26,27,28,29,30]. For instance, the combination of CNT with polymers gives room for the fusion of the elasticity, low density, and simple processing of conventional polymers with the exceptional mechanical, thermal, and electrical properties of the CNT. Moreover, the resultant material could have novel electrical, thermal, and electromagnetic characteristics, thereby further expanding the applications of CNT [31]. However, various levels of toxicity of CNT, depending on several factors, including preparation method, aspect ratio, shape, and surface to volume ratio, among others, have been reported [32,33]. Owing to their unique physicochemical characteristics, the biocompatibility of CNT can be increased via functionalization with different biomolecules, and no severe toxicity or adverse effect for functionalized CNTs were observed, making them leading candidates for several applications in the biomedical field such as tissue engineering applications, gene therapy, drug delivery, and biosensors [34,35].

The preparation and characterization of nanocomposite films based on pullulan have been previously described [36,37]. Additionally, an attempt to obtain nanobiocomposite films based on bacterial cellulose and pullulan [38] and pullulan/nanofibrillated cellulose [39] to obtain sustainable and environmentally friendly materials has been reported. The reinforcement effectiveness and other related characteristics of CNT-filled natural polymers or biopolymers were investigated [40,41,42,43]. Furthermore, the investigation of the combination of CNTs and carbon nanofiber (CNF) has also been reported [44]. Additionally, some pullulan-based nanocomposites have been previously described in the literature [45,46,47].

Herein, we prepared pullulan-based biodegradable conducting multi-nanobiocomposite films by incorporating CNT and TOCNs as fillers. The stable dispersed CNTs by pullulan may have biomedical applications, including tissue engineering and drug delivery, which widen the application area of CNTs. Moreover, owing to the bioactivity of biopolymers, the prepared nanobiocomposite may have effective sensing performance [48] and applicability for electronic packaging [49] and electric equipment [40], among others. TOCNs are considered a promising natural reinforcing agent in polymer nanocomposites, and their surface morphology and physical properties are investigated as a function of TOCNs loading [50,51,52].

In this study, the influence of CNT on the material properties of PULL/TOCNs/CNT nanocomposites was investigated. The TOCNs concentration was kept constant (5 wt.% of the polymer weight), and the influence of the CNT loading was determined. The novelty of this study lies in the transformation of a natural polymer into new advanced materials applicable in various fields such as biomedicine, biosensing, and electronics.

## 2. Materials and Methods

### 2.1. Materials

PULL (Powder form, purity above 90%, molecular weight (Mw) of 20,000) was purchased from Hayashibara Biochemical Laboratories Inc. (Okayama, Japan). Multiwalled CNT (20 nm in diameter, 10 µm in length) was obtained from Antech, Hanoi, Vietnam and aqueous 0.5% *w*/*v* suspension of TOCNs was supplied by Sigma-Aldrich, St. Louis, MO, USA. Doubly distilled water was used as a solvent.

### 2.2. Preparation of the Nanocomposite Film

PULL/TOCNs/CNT nanocomposite film was prepared using the solution casting method. For this purpose, TOCNs, CNT, and PULL were mixed in 100 mL of distilled water using magnetic stirring (400 rpm and 30 °C) and a sonicator. The concentration of PULL and TOCNs was 10 and 5 wt.%, respectively, while the different concentrations of CNT (0, 0.5, 1, 3, and 5 wt.%) were taken separately. PULL was taken according to the solution weight; conversely, TOCNs and CNT were measured depending on the PULL weight. After 10 g of the prepared solution was poured on the polystyrene petri dish (8 × 8 cm^2^), the solution mixture was dried in a drying oven at 35 °C until complete dryness. Finally, the films were peeled out and preserved for characterization. The thickness of the film is summarized in Table 1.

### 2.3. Characterization of the Films

A thickness gauge (Digital Verniercaliper, Hando, Seoul, Korea) was used for measuring film thickness. The surface morphologies of the PULL/TOCNs/CNT composite films were observed using field emission scanning electron microscopy (FE-SEM) (SU8220, Hitachi, Japan) and atomic force microscopy (AFM, Park Systems (NX20), Mannheim, Germany). The preparation of composite films was confirmed using a Fourier-transform infrared (FT-IR) spectrometer (Frontier, Perkin Elmer, Waltham, MA, USA) and X-ray diffraction (XRD) (D/Max–2500, Rigaku, Tokyo, Japan). The thermal and mechanical properties were evaluated with thermogravimetric analysis (TGA) (model Q-50 from TA Instruments, Seoul, Korea) and an Instron 5567 Universal Testing Machine (load cell of 500 Nanda, crosshead speed 20 mm/min), respectively. The mechanical studies followed the ASTM D638-96 type II requirements. For each condition, three samples (20 × 60 mm^2^) were used. The soil burial test was performed according to the literature [53,54], and also the moisture uptake test was performed following the literature [55]. UV/vis-spectroscopy (K Lab Co., Ltd., Optizen 2120UV, Daejeon, Korea) was used to evaluate the film’s light transmittance (T%) in the 200–800 nm wavelength range. The water contact angle was measured using a contact angle meter (Dino-Lite Korea, AM7013MZT, Seoul, Korea) [55]. The short time thermal exposure test was performed by following a reported procedure [56]. The electrical conductivity of the prepared films was calculated from the bulk resistance of the sample with the most precise dimensions. The electrical conductivity (ρ) was calculated using the following Equation:ρ = l/(R × S) = l/(R × t × w);
where S, l, w, and t represent the cross-sectional area, length, width, and thickness of the samples, respectively. After drying, the resistance (R) of the polymer film strips (10 × 30 mm^2^) was measured by a digital multimeter (Tae Kwang Electronics Co., TK-3205, 3204A, Seoul, Korea).

## 3. Results and Discussion

### 3.1. Dispersion of the Nanofillers in the PULL Matrix

#### 3.1.1. FE-SEM and AFM Analysis of the PULL/TOCNs/CNT Nanocomposite Films

Scanning electron micrographs showing the surfaces of the prepared PULL/TOCNs/CNT nanocomposite films are shown in Figure 1. The PULL/TOCNs/CNT 0.5 wt.% film shows a uniform and smooth surface (Figure 1a), whereas the 5% CNT-loaded composite films exhibit an uneven surface (Figure 1b). Increasing the CNT weight proportion interrupts the formation of a smooth surface because CNTs are not soluble or completely dispersible in aqueous biopolymeric solutions, thereby reducing the binding ability of the biopolymer [43]. Additionally, it could result in a decrease in the strength properties, as observed in the mechanical characterizations.

The micromorphology of the PULL composite films was identified by AFM (Figure 2a,b). Compared with the 0.5 wt.% CNT films, more roughness on the surface of 5 wt.% films was observed because of the higher concentration of CNT (Figure 2a,b). The roughness value (Rq) for 0.5 wt.% and 5 wt.% was 0.510 nm and 2.251 nm, respectively. AFM images revealed that the film containing 0.5 wt.% CNT is smoother, probably because of the uniform dispersion of CNT in the PULL/TOCNs matrix, and, on the surface, a lower amount of CNT was exposed to outside, making it rough slightly. The opposite was observed for CNT 5 wt.%. This result is consistent with the findings from the FE-SEM images (Figure 1a,b). Therefore, the composite films with a small amount of CNT provide a favorable surface morphology, which is the basis of the highly stable and efficient composite (Figure 1a and Figure 2a).

#### 3.1.2. FT-IR Analysis

The FT-IR spectra of the neat PULL and PULL/TOCNs/CNT (0–5%) composite films are presented in Figure 3. The characteristic peaks originating from the PULL/TOCNs spectrum show slight movement compared with the neat PULL film, which is probably due to the hydrogen bonds present in the PULL and TOCNs [57,58]. In the PULL/TOCNs/CNT 0.5% spectra, classic peaks ascribed to polysaccharide structures are located at 3000–3600 cm^−1^ (O–H stretching vibrations), 2850–3000 cm^−1^ (CH_2_ and CH stretching vibration), 1300–1500 cm^−1^ (CH/CH_2_ deformation vibration bands), 1000–1260 cm^−1^ (C–O stretching), 932 cm^−1^ (α-(1, 6) glucosides bonds), and 755 cm^−1^ (α-(1, 4) glucosides bonds) [46,58]. Moreover, as the CNTs content increased, the intensity of the characteristic peaks decreased and shifted to a lower value, confirming the interaction between the PULL, TOCNs, and CNT. The –OH stretching vibration band at 3306.98 cm^−1^ in the PULL/TOCNs composite shifts to a lower wavenumber value with increasing CNT contents, confirming the hydrogen bonding between the hydroxyl groups on the CNT surface and the –OH groups in the PULL/TOCNs composite. The aqueous CNT dispersion contains hydroxyl groups on the surface, similar to that reported in previous studies [59].

#### 3.1.3. XRD Data of PULL/TOCNs/CNT Nanocomposite Films

XRD characterization was performed to investigate the crystalline structure of the PULL/TOCNs/CNT films (Figure 4). By increasing the CNT content, the intensity of the characteristic CNT peak increased, which indicates the presence of the reinforcement material, CNT. The peaks associated with the CNT appeared at 2θ of ~26° and 43.4° [60] (as marked by brown shades in Figure 4). The characteristic peaks of the TOCNs were not observed because of the low concentration (5 wt.%) [39,58].

The presence of CNT increases the crystallinity of the PULL/TOCNs composite film, probably because of the occurrence of nucleation polymer crystallization in it. Moreover, the increase of CNT (0.5% to 3%) leads to a slight decrease in the degree of crystallinity, and at 5% CNT, the crystallization rate measured was low. This may be due to the crystal growth being affected with the increase of CNT content. The estimated degree of crystallinity values for the prepared film show the sequence of 52.34%, 54.62%, 59.64%, 59.12%, 59.04%, and 58.62% (within the error range of ca. < ±0.24%) for the PULL, PULL/TOCNs, PULL/TOCNs/CNT 0.5%, PULL/TOCNs/CNT 1%, PULL/TOCNs/CNT 3%, and PULL/TOCNs/CNT 5% composite films, respectively.

### 3.2. Thermal Properties of the PULL/TOCNs/CNT Nanocomposite Films

#### 3.2.1. Thermogravimetric Analysis

The thermal stability of the composite films was investigated by calculating the weight loss of the volatile materials. The TGA diagram of the nanocomposite films is shown in Figure 5a. As can be seen, the thermal degradation process is represented in two steps. The first stage (25–225 °C) of the degradation is because of the presence of low-molecular-weight compounds such as solvent and moisture, among others. The second stage of the thermal degradation process of the films occurs within the temperature range of 275–490 °C. The total weight loss of the thermal degradation process was ~10% for the first stage and 78% for the second stage. The lowest curve represents the PULL/TOCNs/CNT 0.5 wt.%, while the topmost curve represents the composite film with 5 wt.% CNT. The PULL and PULL/TOCNs/CNT (0–1%) curves show a similar trend. Therefore, the higher thermal stability was obtained with the addition of a higher concentration of CNT into the PULL/TOCNs matrix. Moreover, from the differential thermal gravimetric (DTG) curves of the films (Figure 5b), it is evident that the temperature of the maximum weight loss rate gradually increased with the addition of the TOCNs and CNT. However, at 5% CNT, the temperature decreased as a result of a suspected agglomeration. Consequently, the thermal stability was improved with the addition of CNTs into the PULL/TOCNs matrix, and this can be explained by the interaction between the PULL/TOCNs and CNT.

#### 3.2.2. Short Time Thermal Exposure

A short time thermal exposure test was performed to obtain extra information regarding weight loss. Before the thermal exposure, the samples were preserved under standard atmospheric conditions (23 °C and 50% R.H.) for 96 h. Figure 6 shows the weight change of film samples after the thermal exposure test. A continuous weight loss with increasing time and temperature was observed (Figure 6a–d). The lowest weight was found in the case of the CNT 5% at 40 °C and 150 °C (Figure 6d).

### 3.3. Tensile Strength

The mechanical characterization was performed for the neat PULL and PULL/TOCNs nanocomposite films with varying CNT content. Figure 7 and Table 2 show that the addition of different amounts of CNT into the PULL/TOCNs matrix affect the mechanical behavior of the samples. The composite with 0.5 wt.% of CNT loading showed the highest tensile strength (65.41 MPa). However, if the CNT content exceeds 0.5%, the tensile strength decreases. The enhancement of the tensile strength by incorporating the CNT into the polymer is achieved at low CNT content.

Conversely, an increase in the CNT content from 0.5 to 5 wt.% decreased the tensile strength to 34.47 MPa. These findings confirm that the optimal mechanical properties of the PULL/TOCNs/CNT nanocomposites can be enhanced by incorporating nanofiller contents at 0.5 wt.%. At 0.5 wt.%, the CNT is well distributed in the PULL/TOCNs matrix, resulting in improved mechanical properties. In contrast, increasing the filler content causes the agglomeration of the CNT, which inhibits the intended reinforcement. A similar observation has been previously reported [61].

### 3.4. Optical Properties of PULL/TOCNs/CNT Nanocomposite Films

Figure 8a presents the UV transmission spectra obtained for the neat PULL and PULL/TOCNs/CNT (0–5 wt.%) nanocomposite films. The composite films (0.5–5 wt.%) show zero transparency, confirming that the addition of CNT to the PULL/TOCNs matrix makes the film completely opaque.

### 3.5. Conductivity

Like most polymers, PULL is a poor conductor. CNTs are used as conductive fillers in the PULL polymer matrix owing to their low cost and high electrical conductivity [62]. As shown in Figure 8b, the electrical conductivity of the PULL/TOCNs/CNT nanocomposite films was found to increase with the increased CNT content. The electrical conductivity increased from 0.001 to 0.015 S/mm when the CNT content was increased from 0.5 wt.% to 5 wt.%. Hence, the addition of CNT considerably improved the conducting capacity of all investigated films.

### 3.6. Hydrophilic Properties of the PULL-Based Nanocomposite Film

Figure 9 and Table 1 depict the contact angle of the neat PULL and PULL/TOCNs/CNT (0–5 wt.%) films. The surface properties of the films were investigated through the measurement of the contact angle values. A rough surface is necessary for forming superhydrophobic surfaces [63,64]. Furthermore, the addition of the CNT to the PULL/TOCNs matrix enhanced the hydrophobic properties with more extensive contact angle values because of the increase in surface roughness. These results are also consistent with other studies reported in the literature [65].

### 3.7. Moisture Vulnerability

The moisture uptake data plotted in Figure 10 exhibit a downward trend with the addition of CNT loading, and the film containing 5 wt.% shows the lowest value (11.28%), which is ~1% lower than 0.5 wt.% (12.61). This is attributed to the fact that CNT blocks the water molecules distribution at the interface, demonstrating strong filler–matrix adhesion [55]. Moreover, CNT incorporation proved to be an effective strategy to decrease the moisture sensitivity of the PULL/TOCNs composite [58]. The study of moisture absorption is crucial to elucidate the performance of PULL-based composites, as the moisture uptake under either exposure to water or high humidity represents some significant characteristics, including mechanical and physical.

### 3.8. Soil Burial Test

Figure 11 shows the macroscopic images of the films as a function of the time buried in the composite soil. Regardless of the CNTs and TOCNs content of the films, all were completely degraded after 64 days of examination. Significant biodegradation was observed after 16 and 32 days with microorganisms grown all over the film, and the PULL film was entirely degraded. After the soil burial test, the water diffused into the films, causing swelling and accelerating the biodegradation because of the increased microbial growths. The incorporation of the CNTs decreased the moisture absorption rate (Figure 10), inhibited the activity of microorganisms as expected, and reduced the degradation rate. A similar observation was reported by Goodwin et al. [66], who observed that the presence of CNT decreased the biodegradability of polymers.

## 4. Conclusions

A simple solution casting method was developed to prepare thin hybrid films by combining PULL, TOCNs, and CNT. FT-IR results reveal the possibility of interactions between the CNT and PULL, and the TOCNs and PULL matrix. The presence of the reinforcement materials, CNT and TOCNs, was also confirmed using XRD analysis. The maximum degree of crystallinity value was obtained for CNT 0.5 % (59.64%). However, with the increase of CNT (0.5% to 5%), the degree of crystallinity decrease (59.64% to 58.62%) may be due to the crystal growth being affected with the increase of CNT content. With the increasing addition of CNT, the tensile properties, transparency, and flexibility of the composite films were gradually reduced. FE-SEM images reveal a uniform and smooth surface for the lower CNT (0.5%) content film, whereas the 5% CNT-loaded composite films exhibit an uneven surface as CNTs are not completely dispersible in aqueous biopolymeric solutions. Surface smoothness and uniformity of the prepared composite films also decreased radically, as confirmed by AFM. Other important properties of the obtained composite films, such as the thermal and barrier properties, increased as the concentration of the CNT content increased. The differential thermal gravimetric results reveal that the temperature of the maximum weight loss rate decreases at the higher CNT concentration (5%), due to the possible agglomeration. Additionally, it was determined that the PULL/TOCNs/CNT composite film is biodegradable and degrades within 64 days. Moreover, the addition of the CNT to the PULL/TOCNs gives rise to a flexible, electrically conducting hybrid nanobiocomposite film for use in the field of biomedicine, biosensing, and electronics.

## Figures and Tables

**Figure 1 nanomaterials-11-00602-f001:**
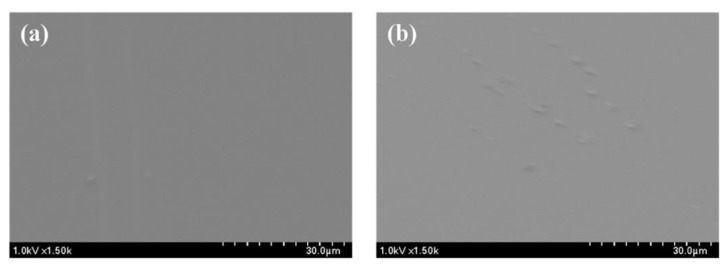
FE-SEM images of the pullulan/TEMPO cellulose nanofibrils (PULL/TOCNs)-based composite film with varying carbon nanotube (CNT) contents of (**a**) 0.5 wt.% and (**b**) 5 wt.%.

**Figure 2 nanomaterials-11-00602-f002:**
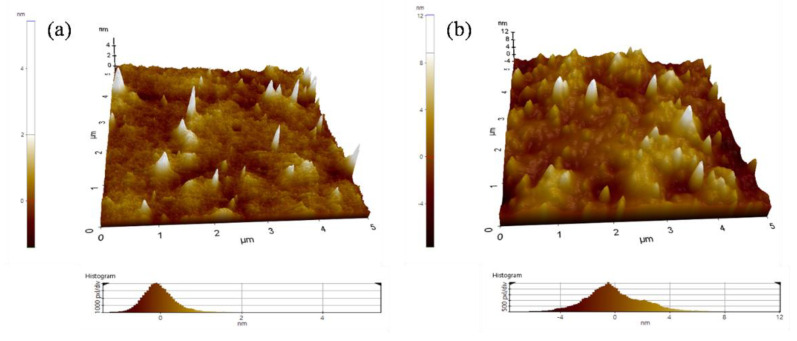
AFM images of the PULL/TOCNs-based composite film with varying CNT contents of (**a**) 0.5 wt.% and (**b**) 5 wt.%.

**Figure 3 nanomaterials-11-00602-f003:**
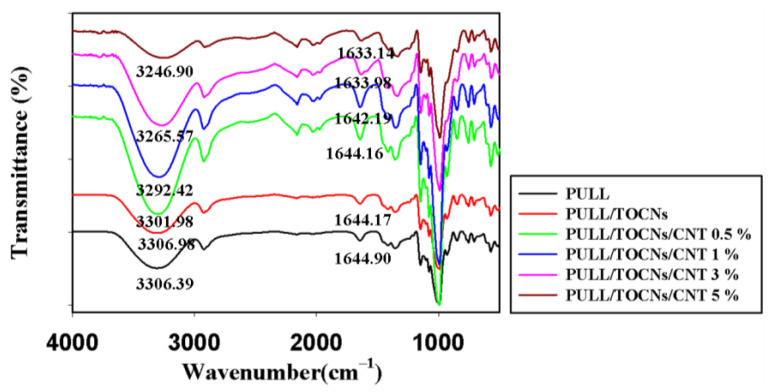
FT-IR data of neat PULL and PULL/TOCNs/CNT (0–5 wt.%) composite films.

**Figure 4 nanomaterials-11-00602-f004:**
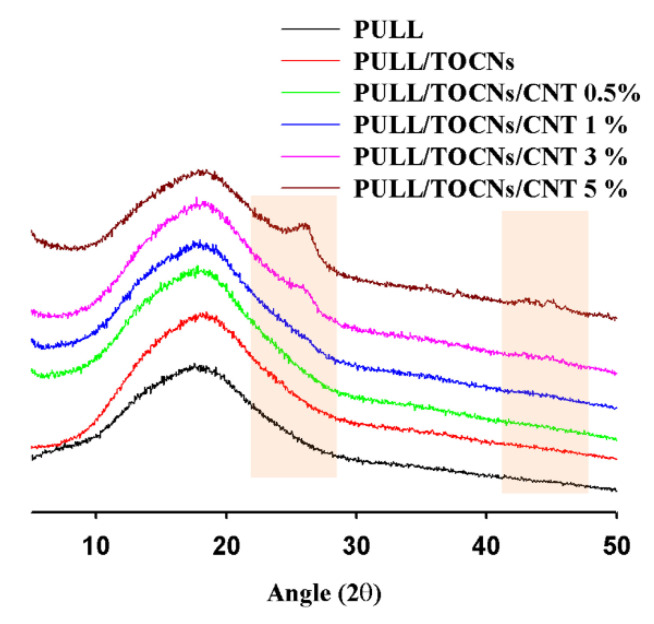
XRD data of the neat PULL and PULL/TOCNs/CNT (0–5 wt.%) composite film.

**Figure 5 nanomaterials-11-00602-f005:**
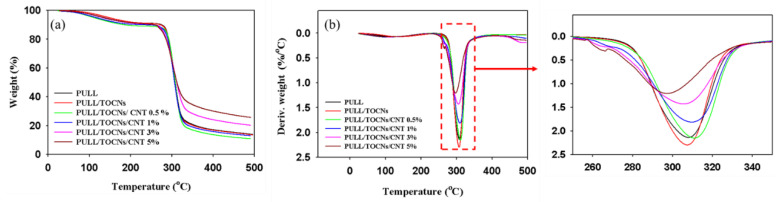
TGA (**a**) and differential thermal gravimetric (DTG) (**b**) results of neat PULL and PULL/TOCNs/CNT (0–5%) composite films.

**Figure 6 nanomaterials-11-00602-f006:**
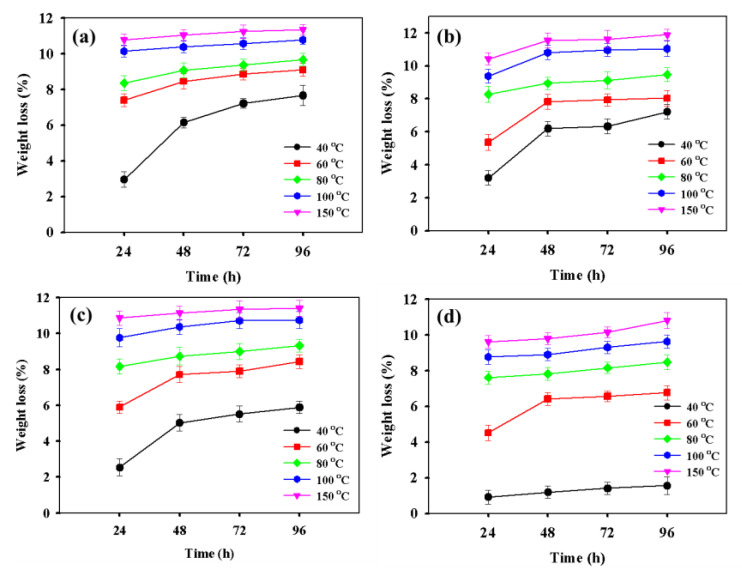
Weight loss data of PULL/TOCNs-based composite film with varying CNT contents of (**a**) 0.5%, (**b**) 1%, (**c**) 3%, and (**d**) 5%.

**Figure 7 nanomaterials-11-00602-f007:**
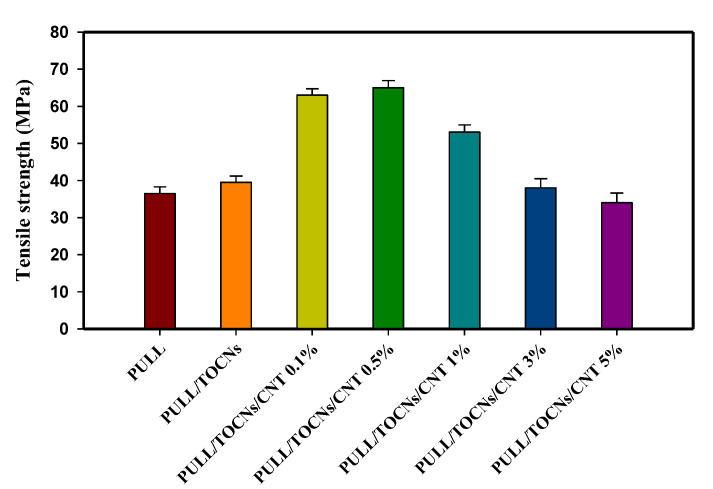
Tensile strength column graph of the neat PULL and PULL/TOCNs/CNT (0–5 wt.%) nanocomposite films.

**Figure 8 nanomaterials-11-00602-f008:**
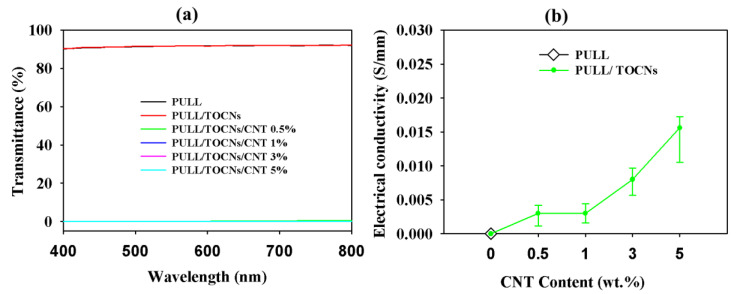
The (**a**) visible light transmittance and (**b**) electrical conductivity of the neat PULL and PULL/TOCNs/CNT nanocomposite films.

**Figure 9 nanomaterials-11-00602-f009:**
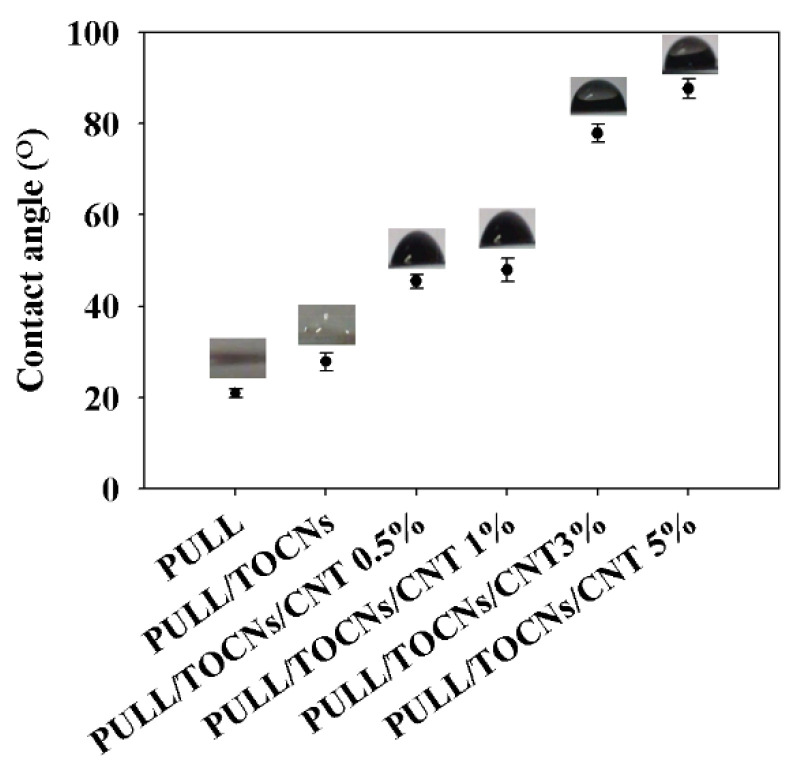
The water contact angle of the PULL/TOCNs/CNT nanocomposite films.

**Figure 10 nanomaterials-11-00602-f010:**
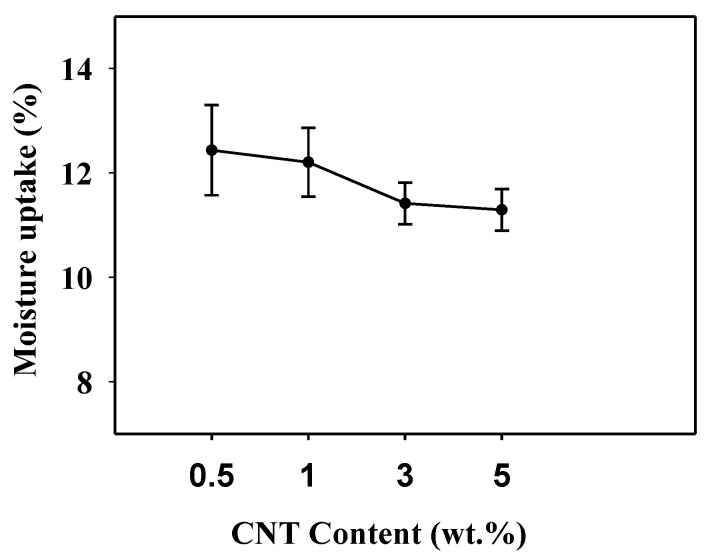
Moisture uptake of PULL/TOCNs/CNT-based nanocomposite films.

**Figure 11 nanomaterials-11-00602-f011:**
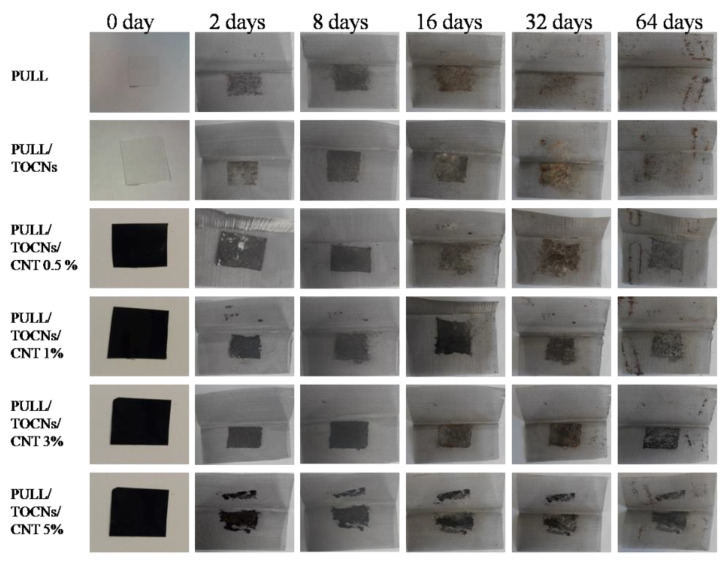
Macroscopic pictures of the biodegradation of different film samples in soil.

**Table 1 nanomaterials-11-00602-t001:** Contact angle and thickness of the PULL/TOCNs/CNT nanocomposite film.

Samples	Contact Angle (°)	Thickness (µm)
PULL	20.95 ± 2	150 ± 2.05
PULL/TOCNs	27.89 ± 1	150 ± 2.07
PULL/TOCNs/CNT 0.5%	45.44 ± 1.50	150 ± 2.27
PULL/TOCNs/CNT 1%	47.99 ± 2.5	150.2 ± 5.34
PULL/TOCNs/CNT 3%	77.90 ± 2.00	150.4 ± 4.44
PULL/TOCNs/CNT 5%	87.69 ± 2.20	150.4 ± 3.32

**Table 2 nanomaterials-11-00602-t002:** Mechanical properties of PULL/TOCNs/CNT nanocomposite films.

Samples	Tensile Strength (MPa)	Elongation at Break (%)
PULL	36.5 ± 1.77	4.55 ± 0.30
PULL/TOCNs	39.5 ± 1.70	4.30 ± 0.20
PULL/TOCNs/CNT 0.5%	65 ± 1.90	7.63 ± 0.55
PULL/TOCNs/CNT 1%	53 ± 2.00	6.33 ± 0.44
PULL/TOCNs/CNT 3%	38 ± 2.49	5.39 ± 0.41
PULL/TOCNs/CNT 5%	34 ± 2.6	5.28 ± 0.66

## Data Availability

The data presented in this study are available on request from the corresponding author.

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
