# Peer review of "Electrically Conducting Pullulan-Based Nanobiocomposites Using Carbon Nanotubes and TEMPO Cellulose Nanofibril"

_nanomaterials, 2021, doi:10.3390/nano11030602_

Round 1
Reviewer 1 Report
I satisfied revised version.
Author Response
Thank you very much for your message of February 10, 2021, regarding our manuscript.
Reviewer 2 Report
The manuscript entitled "Electrically conducting pullulan-based nano-bio-composites using carbon nanotubes and tempo cellulose nanofibril" is a revised version of the paper, revised and resubmitted to Nanomaterials.
The paper deals with the development of multi-functional biodgradable nanocomposites using TEMPO cellulose nanofibrils and carbon nanotubes. The resulting materials possess good mechanical characteristics and excellent electroconductivity, they show thermal and water barrier properties, as well as biodegradability. The resubmitted version was significantly improved by the authors, and the paper deserves to be published upon minor revision (see my comments below):
1) The title of the paper should be corrected (TEMPO instead of tempo, nanofibrils instead of nanofibril).
2) The meaning of the following phrase is unclear: "Therefore, the composite films with a small amount of CNT provide a favorable 148 surface morphology is the basis of the highly stable and efficient composite."
3) The meaning of the following phrase is also unclear: "This may be due to the crystals interfering, affecting the crystals throughout the growth process with the increase of CNT content."
Author Response
Thank you very much for your message of February 10, 2021, regarding our manuscript. We also would like to thank you for sending the reviewer’s comments to us. The revised manuscript and the details of the revisions in the manuscript have been prepared as per your suggestion. Our responses to the reviewer’s and academic editor comments are as follows:
1) The title of the paper should be corrected (TEMPO instead of tempo, nanofibrils instead of nanofibril).
Answer: The title of the paper is corrected as follows “Electrically conducting pullulan-based nano-bio-composites using carbon nanotubes and TEMPO cellulose nanofibrils" [Line 3]
2) The meaning of the following phrase is unclear: "Therefore, the composite films with a small amount of CNT provide a favorable 148 surface morphology is the basis of the highly stable and efficient composite."
Answer: The mentioned line is modified to make it clear.
"Therefore, the composite films with a small amount of CNT provide a favorable surface morphology which is the basis of the highly stable and efficient composite (Figure 1a and Figure 2a)” [Line 148-150]
3) The meaning of the following phrase is also unclear: "This may be due to the crystals interfering, affecting the crystals throughout the growth process with the increase of CNT content."
Answer: The mentioned line is revised as follows:
“This may be due to the crystal growth affected with the increase of CNT content” [Line 181-182]
This manuscript is a resubmission of an earlier submission. The following is a list of the peer review reports and author responses from that submission.
Round 1
Reviewer 1 Report
Comment:
The author described that hybrid bio-nanocomposite films were prepared using a simple solution casting method by incorporating tempo cellulose nanofibrils (TOCNs) and carbon nanotubes (CNTs) into an aqueous solution of pullulan (PULL) and to evaluate its performance of functionality. The reviewer has some questions and comments as blow.
- The authors described that no research study has been reported on pullulan-TOCNs/CNT composite films. Why did no one study about it? What is bottle neck for it and How did you overcome the issue?
- The authors focused on balance of PULL and CNT. However, what effect TOCNs contribute for composite?
- In figure 4, can you discuss and calculate about degree of crystallinity from the results of XRD? The author described changing of intensity indicated the presence of the reinforcement material, CNT. However, decreasing of intensity show the increasing amorphas region in it. This meaning is not the presence of the reinforcement material.
- Regarding figure 8, the author described that the addition of CNT to the PULL/TOCNs matrix makes the film completely in transparent. However, PULL-TOCNs/CNT composite films showed black color in Figure 11.
Author Response
The author described that hybrid bio-nanocomposite films were prepared using a simple solution casting method by incorporating tempo cellulose nanofibrils (TOCNs) and carbon nanotubes (CNTs) into an aqueous solution of pullulan (PULL) and to evaluate its performance of functionality. The reviewer has some questions and comments as blow.
1) The authors described that no research study has been reported on pullulan-TOCNs/CNT composite films. Why did no one study about it? What is bottle neck for it and How did you overcome the issue?
Answer: The mentioned explanation is improved and described the purposes of the study [Line: 62-71].
2) The authors focused on balance of PULL and CNT. However, what effect TOCNs contribute for composite?
Answer: Some references and related description are included in the introduction part of the manuscript about the effect of TOCNs for composite preparation [Line: 79-86].
3) In figure 4, can you discuss and calculate about degree of crystallinity from the results of XRD? The author described changing of intensity indicated the presence of the reinforcement material, CNT. However, decreasing of intensity show the increasing amorphas region in it. This meaning is not the presence of the reinforcement material.
Answer: Results and description regarding the degree of crystallinity are included in the XRD results and discussion part, and the description of the CNT reinforcement has been improved [Line: 175-176, 179-187].
4) Regarding figure 8, the author described that the addition of CNT to the PULL/TOCNs matrix makes the film completely in transparent. However, PULL-TOCNs/CNT composite films showed black color in Figure 11.
Answer: The description regarding transparency is corrected [Line: 245].
Reviewer 2 Report
The paper deals with the fabrication of composite films containing pullulan, nano-cellulose and carbon nanotubes. The results obtained demonstrate enhanced properties of these films including enhaced electrical conductivity and tensile strength. The subject of the paper is within the scope of Nanomaterials.
I have the following comments:
1) No information on CNT toxicity is provided in the Introduction section, while this could affect significantly the use of the composite films.
2) Possible applications of the composite films obtained should be discussed.
3) Possible chemical interactions between the components of the films should be discussed in details.
4) The electrical conductivity graph (Fig. 8b) should be re-plotted to show only the positive values of the conductivity.
5) No data on the surface roughness of the films are provided. These data are also highly important in view of the possible applications of the films.
6) The meaning of some sentences is unclear, e.g. "The peak obtained for PULL/TOCNs..." etc.
7) English should be improved throughout the paper.
Author Response
The paper deals with the fabrication of composite films containing pullulan, nano-cellulose and carbon nanotubes. The results obtained demonstrate enhanced properties of these films including enhaced electrical conductivity and tensile strength. The subject of the paper is within the scope of Nanomaterials.
I have the following comments:
1) No information on CNT toxicity is provided in the Introduction section, while this could affect significantly the use of the composite films.
Answer: Information about CNT toxicity and its effect on the composite is included in the introduction section [Line: 62-70].
2) Possible applications of the composite films obtained should be discussed.
Answer: A description of the possible applications of the prepared composite is added in the introduction section [Line: 79-86].
3) Possible chemical interactions between the components of the films should be discussed in detail.
Answer: Discussion about the chemical interactions between the components of the film has been included in the FTIR results and discussion part [Line: 166-170].
4) The electrical conductivity graph (Fig. 8b) should be re-plotted to show only the positive values of the conductivity.
Answer: The electrical conductivity graph has been upgraded to keep only the positive conductivity values.
5) No data on the surface roughness of the films are provided. These data are also highly important in view of the possible applications of the films.
Answer: Data regarding surface roughness are marked as red color in the AFM results and discussion part and the description is enlarged [Line: 147,151-152].
6) The meaning of some sentences is unclear, e.g. "The peak obtained for PULL/TOCNs..." etc.
Answer: Some sentences are improved to make it more clear [Line:158,160,164-166].
7) English should be improved throughout the paper.
Answer: English is improved all over the paper.
Additional: Ten more new references are added to answer the reviewer questions after reference no 31. So the reference numbers are rearranged after 31.
Round 2
Reviewer 2 Report
The authors have addressed most of my comments. The paper is now suitable for publication, while some English polishing is still required.